# Defining good health and care from the perspective of persons with multimorbidity: results from a qualitative study of focus groups in eight European countries

Fenna R M Leijten,[1] Maaike Hoedemakers,[1] Verena Struckmann,[2] Markus Kraus,[3] Sudeh Cheraghi-Sohi,[4] Antal Zemplényi,[5,6] Rune Ervik,[7] Claudia Vallvé,[8] Mirjana Huić,[9] Thomas Czypionka,[3] Melinde Boland,[1] Maureen P M H Rutten-van Mölken,[1,10] on behalf of the SELFIE consortium

For numbered affiliations see end of article.

**Correspondence to**
Professor Maureen P M H Rutten-van Mölken;
m.rutten@eshpm.eur.nl

## ABSTRACT

**Objectives** The prevalence of multimorbidity is increasing in many Western countries. Persons with multimorbidity often experience a lack of alignment in the care that multiple health and social care organisations provide. As a response, integrated care programmes are appearing. It is a challenge to evaluate these and to choose appropriate outcome measures. Focus groups were held with persons with multimorbidity in eight European countries to better understand what good health and a good care process mean to them and to identify what they find most important in each.

**Methods** In 2016, eight focus groups were organised with persons with multimorbidity in: Austria, Croatia, Germany, Hungary, the Netherlands, Norway, Spain and the UK (total n=58). Each focus group followed the same two-part procedure: (1) defining (A) good health and well-being and (B) a good care process, and (2) group discussion on prioritising the most important concepts derived from part one and from a list extracted from the literature. Inductive and deductive analyses were done.

**Results** Overall, the participants in all focus groups concentrated more on the care process than on health. Persons with multimorbidity defined good health as being able to conduct and plan normal daily activities, having meaningful social relationships and accepting the current situation. Absence of shame, fear and/or stigma, being able to enjoy life and overall psychological well-being were also important facets of good health. Being approached holistically by care professionals was said to be vital to a good care process. Continuity of care and trusting professionals were also described as important. Across countries, little variation in health definitions were found, but variation in defining a good care process was seen.

**Conclusion** A variety of health outcomes that entail well-being, social and psychological facets and especially experience with care outcomes should be included when evaluating integrated care programmes for persons with multimorbidity.

## Strengths and limitations of this study

- ► Focus groups were held in eight European countries among persons with multimorbidity.
- ► The qualitative, focus group approach applied in this study ensures that the perspectives of persons with multimorbidity are incorporated in the discussion on what meaningful evaluation outcomes are.
- ► Only participants in one focus group (in the UK) explicitly named clinical psychological health problems as a comorbidity, which may suggest that perspectives from persons with this type of health problem are under-represented in this study.
- ► Creating an overarching top 10 list of important outcomes in health and care is a summation of a qualitative process, and thus this list should be interpreted cautiously.
- ► Although different countries and multimorbidities were included, the applied qualitative method does not mean that the findings are necessarily transferable to all persons with multimorbidity in all countries.

multimorbidity, that is, the co-occurrence of two or more health problems within one person at one time.[1 2] Persons with multimorbidity therefore often require care from different types of professionals. These professionals may work in different healthcare sectors (eg, primary and secondary care) and may also work in social or community care. It is important that persons with multimorbidity receive well-integrated care, in order to avoid the risk of fragmentation or overlap in the care received and interactions in treatment.[3]

Integrated care is defined as structured efforts to provide coordinated, proactive, person-centred, multidisciplinary care by two or more care professionals that effectively communicate and collaborate. There are different integrated care programmes

## INTRODUCTION

With ageing populations in Western societies, there is an increasing prevalence of

for multimorbidity being implemented across Europe that may offer a solution to the aforementioned risks that this population faces.[4–7] The evidence base of such programmes is still limited, and findings are not yet wholly convincing.[5–7] Increasing this evidence base is important for the durability, wider implementation and more sustainable reimbursement/financing of such programmes. Often, decisions on these matters are informed by economic evaluations in which costs per quality-adjusted life years (QALYs) are calculated. However, it can be questioned whether the current (economic) evaluation framework provides sufficient insight into the broad range of outcomes that such integrated care programmes aim to improve. Integrated care programmes are complex interventions: they consist of various interacting components, target individuals but also groups and organisations, have a variety of intended outcomes, are amendable to tailoring through adaptation and learning feedback loops and their effectiveness is impacted by the behaviour of those delivering and receiving the intervention.[8] Common generic outcomes such as QALYs may not fully capture what these programmes are actually trying to achieve in persons with multimorbidity. Their aims may go beyond the improvement of life expectancy and health-related quality of life and include the improvement of well-being, the maintenance of independence and increasing satisfaction with the care process.

In a time when the scarcity of resources is evident and evidence-based decisions on the spending of these resources are warranted, it is crucial to set up appropriate evaluations that can be used to convince decision makers. Outcomes in the evaluations of complex care programmes often correspond at the higher level to the so-called Triple Aim: improving population health (and well-being), improving the patient's experience with care and reducing cost (growth).[9 10] These higher level outcomes, however, can be interpreted in different ways. Health can be defined as the absence of disease,[11] or a wider perspective can be applied whereby health is seen as the complete physical, mental and social well-being beyond merely the absence of disease.[12] More recent definitions turn health into a more active term, as the ability to adapt[13] and as a 'meta-capability' that can be used to attain human value.[14] Similarly, experience with care can include many different aspects, such as the extent to which care is proactive or person centred (eg, two domains of the Patient Assessment of Chronic Illness Care instrument).[15] Scientific literature used in a review conducted in the context of the Sustainable intEgrated chronic care modeLs for multimorbidity: delivery, FInancing, and performancE (SELFIE) research project, as the current study is (see box 1), provided insight into which outcomes have been used in recent integrated care programme evaluations.[5] We saw that a wide array of indicators, corresponding to the Triple Aim, are being used. This is also reiterated by the findings in the review by Linton *et al*,[16] whereby a total of 99 instruments to assess well-being were found, with a great variety between these.[16]

> ### Box 1 Information on the SELFIE project
>
> Sustainable intEgrated chronic care modeLs for multimorbidity: delivery, FInancing, and performancE (SELFIE) is a Horizon2020-funded EU project that aims to contribute to the improvement of person-centred care for persons with multimorbidity by proposing evidence-based, economically sustainable, integrated care programmes that stimulate cooperation across health and social care and are supported by appropriate financing and payment schemes. More specifically, SELFIE aims to:
> ► Develop a taxonomy of promising integrated care programmes for persons with multimorbidity.
> ► Provide evidence-based advice on matching financing/payment schemes with adequate incentives to implement integrated care.
> ► Provide empirical evidence of the impact of promising integrated care on a wide range of outcomes using Multi-Criteria Decision Analysis (MCDA).
> ► Develop implementation and change strategies tailored to different care settings and contexts in Europe, especially Central and Eastern Europe.
>
> The SELFIE consortium includes eight organisations in the following countries: the Netherlands (coordinator) (NL), Austria (AT), Croatia (HR), Germany (DE), Hungary (HU), Norway (NO), Spain (ES) and the UK. Each country has a unique healthcare and social care system that varies in the extent to which healthcare and social care are integrated and that vary in how the financing is arranged. On the SELFIE website (www.selfie2020.eu), a macro-level description of the systems can be found that provides background context.
>
> Seventeen promising integrated care programmes for persons with multimorbidity across these eight countries are being evaluated in SELFIE. The evaluations apply MCDA and each use a common set of core outcomes as well as programme-type specific outcomes. The latter depend on whether a programme is (1) a population health management programme, (2) a programme targeting frail elderly, (3) a programme targeting persons with problems in multiple life domains or (4) an oncology or palliative care programme.
> Grant agreement no. 6 34 288.

Different perspectives and approaches can be taken when operationalising the Triple Aim and meaningful outcomes of integrated care programmes. In the current study, we aim to better understand what good health or well-being and a good care process mean from the perspective of persons with multimorbidity and to identify what they find most important in each. We hereby thus focus on two of the three 'Triple Aims'. We look at this from a cross-country perspective by conducting focus groups in eight European countries involved in the SELFIE research project on integrated care in multimorbidity: the Netherlands, Austria, Croatia, Germany, Hungary, Norway, Spain and the UK (see box 1). Using a qualitative focus group approach encourages interaction between persons to take place that may allow for novel concepts and themes to arise.

## METHODS

Focus groups were chosen because we were interested in the perspectives of persons with multimorbidity themselves when it comes to health/well-being and care. The

qualitative focus group methodology allows for novel concepts to arise, and interaction between persons can strengthen this process. The Consolidated criteria for Reporting Qualitative research checklist was used to structure this manuscript.[17] Eight focus groups were conducted with persons with multimorbidity, one in each SELFIE partner country: the Netherlands (Rotterdam), Austria (Vienna), Croatia (Zagreb), Germany (Berlin), Hungary (Budapest), Norway (Bergen), Spain (Barcelona) and the UK (Manchester). These countries have a long history of universal healthcare systems that are either tax funded or funded through insurance premiums. However, they each have a unique healthcare and social care provider system in which persons with multimorbidity might face different challenges. For more information on the organisation of care in each context, see the macro-level descriptions of each country in the 'thick description' reports available on the SELFIE website (www.selfie2020.eu). The Dutch SELFIE team provided all other partners with a protocol on how to conduct and report on the focus groups, held a teleconference with each partner to discuss the protocol and provided additional support throughout the process. All focus groups were organised between spring and fall of 2016.

### Recruitment and patient and public involvement

The goal was to recruit 6–8 persons for each focus group, with a mix of gender, age distribution and types of multimorbidity (≥2 health and/or social problems) being desired. Different recruitment strategies were applied: participants were recruited via patient organisations (Austria (AT), Germany (DE), Croatia (HR), the Netherlands (NL) and Norway (NO)), medical professional organisations (HR), self-help groups (AT and DE), medical centres (AT, Spain (ES) and Hungary (HU)), non-profit care organisations (eg, Red Cross) (AT) and patient and public involvement groups (UK). We specified that we were searching for persons with multiple health and/or social problems. Participants were also recruited via SELFIE national Stakeholder Advisory Boards (SAB) (AT, DE, HU, NL and NO). Each country has a SAB that reflects on SELFIE findings and consists of five stakeholder groups (5Ps): patients (persons with multimorbidity), partners (informal caregivers), professionals, payers and policy makers. At the time that the focus groups were being organised, a SAB meeting had just taken place in each country, with two to four persons with multimorbidity present. These persons were reached out to for the focus groups, and we tried to snowball via their networks, for example, also via patient organisations.

Participants were reimbursed for their travel costs, and in some countries a gratuity was made available either as a token of appreciation or to aid recruitment (AT, DE and NL). As in most cases no verbatim transcripts were made, these were not returned to participants. However, in some cases, participants were emailed after the meeting to thank them for their participation (DE, ES, HU, NL, NO and UK) and also in some cases were sent notes (NL).

About 1 year after the focus groups, all participants were sent an update on the overall focus group results across countries. The current publication is also to be shared with participants.

### Procedure

An extensive protocol was developed (see online supplementary file box S1) that was followed in each focus group. The first focus group was held by the Dutch team, who added 'lessons learnt' to the protocol to aid the subsequent focus groups. Each country made a protocol in their own language to use during the actual focus group, which was held in their respective language. The focus groups were all held in a meeting room in an office, university or clinical setting, with two to four researchers from the respective country present, one being the chair and at least one taking extensive notes throughout the meeting. At least one of these researchers is a coauthor on this publication (FL, VS, MK, SC-S, AZ, RE, CV, MiH, MB and MPMHR-vM). The focus groups were also recorded. These researchers had a minimum of a bachelor's degree and experience/training in patient contact and/or qualitative research.

The focus groups consisted of an introduction and two main parts: (1) defining (A) good health and well-being and (B) a good care process, and (2) discussion on most important concepts and creating 'top 10' lists. The planned duration of the focus groups was 2 hours.

At the onset, the researchers welcomed everyone and introduced themselves, as only some researchers present had already been in touch with participants via telephone or email beforehand. The researchers stated their names, current position and background expertise/occupation. Following this, in the introduction, the researchers stated the purpose of the focus group: to discuss what is important from the perspective of an individual with multiple health and/or social problems when it comes to health and care. The agenda of the focus group (introduction, part 1 and part 2) was described. Next, the 'rules' of the focus group were introduced (see online supplementary file box S1) (eg, respectful interaction and phones off). Participants signed the informed consent form (see ethics statement below). The recorder was then turned on, and participants were asked to introduce themselves and briefly describe their multimorbidities.

In part one of the focus group, participants were asked to define good health/well-being and a good care process. First, a discussion about good health/well-being was held, followed by a discussion about a good care process. We started by asking participants to complete the sentence 'For me, being in great health means…' and 'I'd be really satisfied with all of the care/the overall care that I receive, if…' Answers were discussed and written on flip-over boards. During this discussion, a researcher asked triggering questions when needed and tried to focus the discussion on either health/well-being or care. After this, a researcher went through the statements on the

flip-over board and the group tried to move from specific examples to general outcomes. These more general outcomes were highlighted/marked on the flip-overs and written on cards. Again, a researcher needed to ask thought-provoking questions: what do these original statements really boil down to? How can you generalise this so that it is applicable for others in the room? Throughout both steps, a researcher also could pose the question as to how this is especially relevant or different for multimorbidity.

After part one, there was a 15–30 min break. During the break, the researchers arranged the cards with concepts, or outcomes, mentioned during part one (eg, on tables or white/magnet boards). They also sorted through the a priori made cards from the literature and added those *not* mentioned during part one to the display. These cards were made a priori by the Dutch team and distributed to each organiser. The outcomes on the a priori made cards stemmed from publications on integrated care programmes for multimorbidity that were included in a large scientific literature review that was also conducted in the SELFIE project.[5] All outcomes included in those integrated care evaluation publications were ordered according to the Triple Aim and, where possible, overlap was removed. This resulted in a list of 77 potentially relevant outcomes (51 health/well-being, 22 experience and 4 cost) (see online supplementary file box S2). Each focus group organiser had been asked to translate these outcomes to their respective languages and write them on cards, using different colours for each Triple Aim. The purpose of including also these outcomes in the second part of the focus groups was to provide participants with a large and 'complete' (on the basis of the available literature) overview of possible outcomes.

In part two, the researchers briefly explained to the group that the cards from part one were now on display and that they also added novel cards on the basis of findings in the literature. These new concepts were mentioned one by one and, where unclear, explained. Participants were asked to look at all the concepts. They were asked to write down the 10 concepts that were most important to them on a sheet of paper. Hereafter, a discussion was opened as to what was on everyone's 'top 10' list.

## Analyses

All focus groups were recorded and extensive notes were made, pictures of flip-overs and cards were made and the top 10 lists were collected. In two cases, verbatim transcriptions were also made (HR and NO). Reports were made in English by each SELFIE partner on their respective focus group, following a predefined template: structure of the focus group, recruitment, participant characteristics, reflection on the process, findings part one (concepts reflecting good health/well-being and a good care process), findings part two (summation of top 10 lists) and conclusion/discussion.

Three analytical steps can be distinguished. First, throughout the reporting and summarising done for each focus group, the researchers present at the focus group reflected on their respective focus group; at least one of these researchers is a coauthor. Second, these reports were analysed independently by the first, second and second-to-last author from the Dutch team. During this analysis step, themes and priorities, corresponding to part one and two, were extracted from each focus group and these were compared across focus groups. Third, these themes and priorities were discussed among all coauthors, thus including at least one of author of each initial report who was present at the focus group. The analysis of part one was done applying an inductive (bottom-up) approach, whereby different specific concepts mentioned by participants were analysed by describing these, looking at their overlap and differences and when possible clustering these into more general concepts. By deciding upfront that the procedure and our findings would be structured according to the Triple Aim, we also applied a deductive (top-down) approach. For part two, further clustering was done to present the findings in a clearer way, and a summation was used to identify concepts that were found most important. No specific qualitative analysis software was used in analysing the data. The goal from the outset was to conduct one focus group per country, thus data saturation was not discussed a priori.

## Ethics statement

This study was conducted in line with the Declaration of Helsinki. According to the laws of most participating countries (AT, HR, DE, HU, NL and NO), this type of study is exempt from formal ethical approval by a Medical Ethics Committee. In line with their national laws, two countries (ES and UK) obtained ethical approval. This is not a study in which participants were allocated to treatment groups or underwent any treatment; participants were not recruited as 'patients' but merely as persons living in the community with multimorbidity. Participation was completely voluntary. All recordings, transcripts and analyses were treated anonymously. All focus group participants signed an informed consent form; they received this form prior to their attendance to the focus group and at the start of the focus group had time to ask questions and sign the form. This form was developed by the Dutch team on the basis of the WHO template for informed consent for qualitative research[18] and translated by each country into their respective language. The informed consent consisted of information on the study, the purpose of the research, type of research, participant selection, voluntary participation, procedure, duration, potential risks, benefits, reimbursements, confidentiality, sharing of the results, right to refuse/withdraw and contact information.

**Table 1** Focus group participant characteristics

| | # (M/F) | Mean age (min–max) | Mean number of morbidities |
|---|---|---|---|
| Austria (AT) | 7 (5/2) | 72.9 (62–84) | 2.3 |
| Croatia (HR) | 7 (4/3) | 51.7 (31–69) | 3 |
| Germany (DE) | 12 (4/8) | 62.4 (37–78) | 3.9 |
| Hungary (HU) | 6 (1/5) | 64.5 (47–78) | 3 |
| Netherlands (NL) | 7 (5/2) | 66.3 (53–75) | 5 |
| Norway (NO) | 7 (2/5) | 65.4 (42–76) | 3.1 |
| Spain (ES) | 6 (5/1) | 70.2 (60–81) | 2.7 |
| UK | 6 (4/2) | 68.8 (58–86) | 4 |
| Total | 58 (30/28) | 65.3 (37–86) | 3.4 |

## RESULTS

### Participant and focus group characteristics

In total, 58 persons participated across the eight focus groups. Only in some cases specific persons did not participate, this was because we could not get in touch with these persons, they had a conflicting agenda/schedule (eg, holiday), were too busy or in some cases had to cancel last minute due to illness or last minute (healthcare) appointments.

In all focus groups six or seven persons participated, except in Germany, where 12 persons participated. During part two of the focus group in Germany, two subgroups were created to ease discussion. The mean age

per focus group is presented in table 1. The overall mean age was 65 years (range 31–86).

During the focus groups, most frequently three researchers (min 2, max 4) were present, whereby one led (part of) the discussion and one took notes (total researchers present=20, M/F=4/16). The focus groups took 2.5 hours on average (min 2, max 3).

The majority of persons named three morbidities during the introduction of the focus groups (table 1). Health problems were categorised according to the WHO 2010 International Classification of Diseases.[19] For an overview of the specific morbidities mentioned by each participant at the start of the focus groups, see table 2. Across countries, diseases of the circulatory system, musculoskeletal system and endocrine disorders were most common morbidities among participants (table 2). Specific examples of frequently mentioned diseases were high blood pressure, rheumatism, arthritis and diabetes mellitus. It is of note that only in the UK mental health problems were explicitly named at the start of the focus groups (table 2).

### Defining health and care (part 1)

#### Defining health

Across all focus groups, the idea of good health being the ability to do '*normal' daily activities* was mentioned. This for example included activities such as going outside and undertaking activities (ES), being able to use a computer (UK) and doing physical activities within realistic reach:

**Table 2** Morbidities of participants per country categorised according to the ICD-10

| | Total | Austria | Croatia | Germany | Hungary | Netherlands | Norway | Spain | UK |
|---|---|---|---|---|---|---|---|---|---|
| Neoplasms | 8 | 3 | 1 | 2 | | 1 | | 1 | |
| Diseases of the blood and disorders involving the immune mechanism | 3 | | 2 | | | | 1 | | |
| Endocrine, nutritional and metabolic disease | 31 | 2 | 1 | 6 | 5 | 5 | 4 | 3 | 5 |
| Mental and behavioural disorders | 6 | | | | | | | | 6 |
| Diseases of the nervous system | 20 | 1 | 2 | 12 | 2 | | 1 | | 2 |
| Diseases of the eye and adnexa | 7 | | | | 1 | 6 | | | |
| Diseases of the ear and mastoid process | 5 | | | 3 | | | 1 | | 1 |
| Diseases of the circulatory system | 44 | 1 | 5 | 12 | 8 | 7 | 4 | 4 | 3 |
| Diseases of the respiratory system | 17 | | 1 | | | 6 | 4 | 6 | |
| Diseases of the digestive system | 10 | 1 | 1 | 1 | 1 | 2 | 1 | 1 | 2 |
| Diseases of the skin and subcutaneous tissue | 3 | 1 | 1 | | | | | | 1 |
| Diseases of the musculoskeletal system and connective tissue | 36 | 6 | 6 | 9 | 1 | 4 | 5 | 1 | 4 |
| Diseases of the genitourinary system | 4 | | | | | 4 | | | |
| Congenital malformations, chromosomal abnormalities | 1 | 1 | | | | | | | |
| Symptoms, signs, abnormal clinical and lab findings, not else classified | 2 | | | 2 | | | | | |
| Injury, poisoning and other consequences of external causes | 1 | | | | | | 1 | | |

ICD-10, International Statistical Classification of Diseases and Related Health Problems, 10th revision.

… Also being active is necessary for well-being. I was always very athletic nowadays only limited, but it works if you adapt your activities to your physical ability. (P2, AT)

*Response*: That's true, though I cannot climb mountains anymore, but I can walk through different parks and I can still use the stairways. (P7, AT)

In three focus groups, it was also said that especially in the case of multimorbidity, it is not only about doing such daily activities, but being able *to plan* them and *structure* them yourself (DE, NO and UK). This is related to having realistic expectations about what one can do and relates to acceptance. *Acceptance* was mentioned in terms of self-acceptance and acceptance by others (AT, DE, HU and NL). For example:

Part of it is acceptance, I can do what I can do, and I should leave the other things. (P4, NL)

How to handle the pain and the disease plays an important role. Integration of your diseases in your daily activities, accepting the pain and especially not feeling bad all the time although the diseases are permanent is a big step. (P11, DE)

For me, I even feel good, if no additional things come. In my case, all of my conditions and problems that I was dealing with are considered as end-stage or final stage, and this is accepted. Therefore if my condition is not worsening, then I am fine. (P5, HU)

Daily activities are also linked to the desire to maintain social relationships and participation in society (AT, DE, ES, HR, HU, NL and NO).

I want to be consciously active in society. With everything, doing what I want to do, being useful for others. (P7, NL)

For me, good health means being psychosocially active… (P1; HR) (ie, having social relationships that are meaningful, being an active participant in everyday life)

In many countries, good health was also associated with the *absence of shame, discrimination, fear and/or stigma* (ES, HR, HU, NL, NO and UK). These feelings could relate to the wider public, the patient–provider or the personal realm. For example, in Spain, the examples of fear of walking alone and being vulnerable or shame of being seen with medical apparatuses such as an oxygen machine were mentioned. In Hungary, shame was mentioned in the patient–provider relation, for example, not being able to keep to a diet. In Croatia, these concepts were summarised as personal vulnerability. Fear of the future and not knowing how the disease trajectory will go was also discussed (HU, NL and NO). Doubts and worries about sharing updates on one's diagnoses with family and the impact and burden that might have was mentioned (AT). In the Spanish focus group, the discussion on fear and shame was heated, as persons dealt and coped with

this issue differently. This relates to a point that also came up, of accepting the health problems and *coping*, being resilient, managing and having responsibility for the diseases themselves, which may be perceived to be especially difficult in the case of multimorbidity (DE, HU, NL, NO and UK).

Oh yeah, I think if you have multiple (diseases) it just adds to the whole workload really, how to cope with different things, oh yeah. (P1, UK)

The general idea of good health being defined as *feeling safe* also was discussed in various focus groups (ES, HU, NL, NO and UK). This can be seen as the result of being absent from the aforementioned negative states (eg, fear). Feeling safe was also discussed in terms of trusting professionals (discussed more below). In Norway, feeling safe also extended to the economic realm:

… fear, fear for one day having to give up your work for example … if I don't manage any more, if I am unable to work anymore, then we do not have the economic resources to live here anymore, this was in my thoughts when the illness hits me and then fear… (P2, NO)

Lastly, during the open discussion in part one, participants mentioned having a positive frame of mind, being able to enjoy life and the importance of *psychological well-being* was mentioned in all focus groups.

I think psychological problems ought to be mentioned. Many people with chronic illnesses are also struggling psychologically. It could be because you have bad conscience, because you are dreading something, because you do not know if you will manage and suffer from performance anxiety. (P1, NO)

I want to enjoy my life, even though I've these diseases. That means, just being full of life. (P4, AT)

### Defining care

When considering the wide array of themes discussed when defining health/well-being, it is not surprising that good care was defined by many persons as being approached as a whole person and *being treated holistically* (AT, NO and UK).

I wish that people treat me in a respectful manner, because it's true, I am sick, but the disease is not me. I don't want to be reduced to my diseases. (P2, AT)

So I would have wished for a doctor that, to put it this way, had the overview of the whole human being, that he shouldn't treat a heart disease just in isolation, you have another disease, and a third… (P2, NO)

Persons explained that being approached holistically also means receiving *holistic support*, including informal caregiver support (HR, NL, NO and UK), good information provision and especially *emotional and psychological support*. Participants mentioned that support should take the form of more extensive, easily accessible and

timely information on the health problems at hand and medication, and also psychological support, support with self-management and self-help groups for example (AT, DE, ES, HR, HU and NL). Concerning emotional and psychological support specifically, this is needed even when this is not the 'main' problem at hand (AT, ES, HR, NL and UK). For example, in the Netherlands, participants mentioned that in the case of multimorbidity, it takes time to accept the new health problems that arise on top of existing health problems and that there should be support for this adjustment period. In the UK, a disconnection between the patient and provider's priorities was seen, as it was mentioned that the healthcare professionals do not focus on the mental health issues, when they should:

> Yeah, for me I'd like the mental health to be bigger… for me all my healthcare professionals see the (physical disease) as the big thing with me, I don't, I see my depression as the big thing because that's what affects me day to day. (P4, UK)

> What I see is that there's no psychological aid. When you are told you are a chronic patient and you have to take a drug all your life, and that this is for all your life, some people are depressed, needs psychological aid… (P1, ES)

A wider theme discussed was *trust in professionals and the system*. This is in part linked to the emotional and psychological support, whereby participants felt that two-way trust is needed in the relationship between the patient and professional so that psychological issues can be discussed, the professional really listens and so that they can embark on the care process together (AT, NL, NO and UK). Related to this, respectful interaction between professional and patient was also often mentioned (AT, DE, NL, NO, UK and HU). Specifically in the UK, the direction of trust was mentioned with participants saying that they needed to be 'believed' by the professionals (P2 and P4, UK).

Trust also pertained to being able to rely on the professional in being able to help, based on their skills and knowledge (DE, ES, HR, NL, NO and UK). Participants spoke of having a 'prepared' care professional to talk to and trusting their expertise and education.

> Yeah, but there are difficulties with medication when you've got multiple things… (P5, UK)

> *Response*: Oh yeah, but you'd expect your GPs to keep on top of that, it is debatable as to whether that actually happens or not. (P1, UK)

In HU, DE and NL, the trust also pertained to the professionals in general:

> Trust in the medical world, well I'm sceptical about that, a lot of things happen around you, that you think, did they not see that?… Yeah, then you lose faith/trust because of the things that happen to you… (P2, NL)

If I visit my doctor, I wish I could reproduce what he is doing there, which services are provided and I would like to sign for them. I want to be more informed about what is done. This way, I often cannot trust my physicians. (P8, DE)

As may be especially relevant in multimorbidity, issues surrounding *continuity of care* were mentioned in all focus groups. This pertained to clear responsibilities, a clear contact point, transfer and 'after care', good communication and good collaboration and teamwork. These points about continuity often also related to *sharing information or medical records* between providers and organisations (DE, ES, HR, HU, NO and NL).

> The problem is the coordination between the primary health centres with the reference hospitals. The doctor at the hospital should see all the information, and the family doctor as well… (P5, ES)

> I would expect that all the institutions and all the GPs in this city would be connected to the same electronical [IT] system. In this case I would not need to go with all those stacks of papers when I see a doctor. (…) There should be a system which can be seen by everyone and not only by some particular segments of the care. (P1, HU)

### Cross-country comparisons in defining good health and care (part 1)

Little variation was seen between countries when defining health (part 1). Themes surrounding the definition of a good care process differed more between countries. Although aspects of access and availability were mentioned in all countries, this concept took a different form between the countries. In some cases, this pertained to timely care in terms of waiting lists and enough time (ES, HU and NL). In Hungary, however, this specifically had to do with the lack of information on the waiting times. Persons emphasised that waiting times were more acceptable if they knew in advance how long they had to wait.

> The worst thing is waiting… They could calculate an order with some gap in between. There are problems with the information sharing. (P6, HU)

Access in the form of time also referred to professionals having enough time for persons with multimorbidity in DE, HU and NO.

Availability also had different definitions per country; in HR, DE and the UK, this was treatment, care and provider availability and the freedom to choose between them. In NO, ES and the UK, this was also pertained to geographical availability and geographical access.

Several concepts were only brought up in part one by certain countries. Namely, bureaucracy or the reduction of the burden thereof was mentioned in DE and the NL, for example, with regard to care services that fall within our outside of the insurance package.

There are persons who are physically not able to walk to their physiotherapist anymore. In fact they are in need of transport, but you have to apply for transportation for every single therapeutic unit. These are very difficult circumstances due to too much bureaucracy. This is disastrous… (P9, DE)

In Austria participants discussed the need for more self-help and support groups. In contrast, in the Netherlands, this was mentioned as a positive aspect already present and available. Participants in Austria were aware that these things exist in other countries and felt it was missing in their own context.

Lastly, physical surroundings in care provision were mentioned in Hungary that were related to cleanliness, enough personal space and enough seats.

Because if we consider our life or our homes, we do not like if it is dirty and messy. In a good facility there should be cleanliness, order and discipline. (P2, HU)

At the … [department] patients need to wait in a very narrow corridor with one row of chairs. There are about three times more patients than chairs. (P5, HU)

Furthermore, in Hungary, systematically organised operating procedures in care was an important requirement mentioned by multiple participants. The lack of organised procedures is exemplified by the fact that the care process in hospitals is sometimes unreasonably long, the staff is rushed, there are no clear instructions on the next steps of the procedure, there are redundant diagnostic tests and information is not shared appropriately among professionals.

### Most important concepts (part 2)

In all focus groups, in part 2, participants were asked to make a top 10 list of most important concepts. These concepts were those identified in part 1 and supplemented with 'missing' concepts from the literature (see online supplementary file box S2). These top 10 lists were discussed. There was large variation within focus groups (ie, countries) as to what concepts were deemed most important and on participants' top-10 lists. In online supplementary file box S3, all concepts written on persons' top 10 lists are presented, including those from the literature and 'novel' concepts derived during part 1 of the focus groups.

In table 3 below, an overview of the most frequently mentioned concepts is presented. These are concepts written on the top 10 lists of at least 10 persons (out of the 58 participants in total) across all countries. The following health and well-being concepts were most frequently noted by participants on their top 10 lists: social relationships, a positive frame of mind or resilience, enjoyment of life and maintaining independence. A positive frame of mind or resilience was mentioned by at least one person in each country's focus group. Many facets pertaining to good interactions between care professionals and

persons with multimorbidity were on the top 10 lists of participants, such as good communication, shared decision making and respect. Furthermore, individualised care planning and a holistic assessment and understanding of the problems at hand were aspects of the care process also frequently on the top 10 lists. Lastly, proactive, prevention-oriented care was found important by many participants. As can be seen, there is great variety in concepts that participants put on their top 10 lists, both within and across countries. Furthermore, some phrases and words are quite specific, whereas others are broad; participants were free to determine at what conceptual level they wrote their top 10 outcomes.

## DISCUSSION
### Main findings

Participants defined good health and well-being in terms of being able to conduct 'normal' daily activities, being able to plan and structure these and having social relationships and participating in society. Acceptance by oneself and by others and coping with one's current health situation were deemed aspects of good health. Absence of shame and discrimination, fear and/or stigma from the public, care professionals and oneself, and on the other end, feeling safe and psychological well-being were also facets of good health. Social relationships, resilience, enjoyment of life and maintaining independence were considered the *most* important aspects of good health across participants from all countries. A good care process was defined as one whereby persons are approached and supported holistically, with specific attention for emotional and psychological support, there is confidence and trust in professionals and the system, continuity of care is guaranteed and where information is shared and accessible within a reasonable time. Concepts deemed *most* important were good communication, shared decision making and respect between care professional and the person with multimorbidity, as well as individualised care planning and proactive, prevention-oriented care. Little cross-country variation in health themes were found, however, in defining care differences did exist, for example, in terms of the exact type of access referred to (eg, geographical distance to care providers and timely access).

### Interpreting findings

Several themes brought up during the focus groups were explicitly mentioned to be more relevant for persons with multimorbidity; this was especially the case when defining care. One such concept was enough time: in Norway, persons mentioned that it is difficult when their issues are not immediately visible for a care professional and that they need time to explain the multiple problems at hand. Moreover, confidence in professionals' skills are related to their needing to address or at least be aware of multiple problems and often multiple medications, some of which may go beyond their specialist area of expertise.

**Table 3**  Most frequently listed outcomes on the 'top 10' lists of focus group participants across the eight focus groups

| | #/58 | Austria | Croatia | Germany* | Hungary | Netherlands | Norway | Spain | UK |
|---|---|---|---|---|---|---|---|---|---|
| **Health and well-being** | | | | | | | | | |
| Energy and fatigue | 12 | 4/7 | | | 1/6 | 4/7 | 2/7 | | 1/6 |
| Feeling safe | 10 | | 1/7 | | 3/6 | 3/7 | | 3/6 | |
| Cognitive functioning | 12 | 4/7 | | 1/6 | | 2/7 | 1/7 | 3/6 | 1/6 |
| Maintaining independence | 16 | | 2/7 | 5/6 | 1/6 | 1/7 | | 1/6 | 1/6 |
| Enjoyment of life | 16 | 2/7 | | 4/6 | 1/6 | 3/7 | 3/7 | | 3/6 |
| Positive frame of mind, resilience | 16 | 2/7 | 3/7 | 2/6 | 2/6 | 1/7 | 2/7 | 2/6 | 2/6 |
| Self-esteem | 11 | 1/7 | 1/7 | 3/6 | | 3/7 | 2/7 | | 1/6 |
| Social relationships | 17 | 3/7 | | 5/6 | 2/6 | 4/7 | | 2/6 | 1/6 |
| Societal participation | 12 | 1/7 | 3/7 | 3/6 | | 3/7 | | 1/6 | 1/6 |
| **Experience** | | | | | | | | | |
| Individualised care planning/tailored care | 13 | 1/7 | | 4/6 | 1/6 | 2/7 | 4/7 | | 1/6 |
| Holistic assessment/understanding | 11 | 1/7 | 1/7 | 4/6 | 2/6 | 2/7 | | | 1/6 |
| Good communication between provider and patient | 14 | 1/7 | 1/7 | 2/6 | | 2/7 | 1/7 | 3/6 | 4/6 |
| Shared decision-making professional–patient | 13 | 2/7 | | 4/6 | 1/6 | 1/7 | 1/7 | | 4/6 |
| Respectful interaction between professional and patient | 12 | 2/7 | 1/7 | 5/6 | | | | | 4/6 |
| Shared information between professionals | 10 | 2/7 | 1/7 | 1/6 | 2/6 | 2/7 | | 1/6 | 1/6 |
| Team work between professionals and providers | 10 | | 1/7 | | 2/6 | | 2/7 | 2/6 | 3/6 |
| Confidence in knowledge and skills in professionals | 10 | 1/7 | 2/7 | 3/6 | 2/6 | 1/7 | | 1/6 | |
| Proactive, prevention-oriented care | 13 | | 1/7 | 1/6 | 4/6 | 4/7 | | | 3/6 |

*Group split into two, so each time # out of six persons, each group discussed either health/well-being and costs or care and costs.

Considering the different combinations of morbidities represented in the focus groups and the fact that participants are living in different healthcare and social care contexts, it is interesting to note that regardless of whether the main care professional came from primary, secondary or social care, these aspects relating to respect, providing emotional support and expertise were important for many participants. This might be central to the fact that persons with multimorbidity deal with multiple care professionals who need to have attention for the person as a whole, considering all their morbidities and the emotional burden that comes alongside these. In line with this, well-coordinated and smooth transitions (ie, continuity of care) are particularly important for persons with multimorbidity, who often cross provider, organisation and sector boundaries throughout their care trajectories. In the realm of continuity of care, in many focus groups, sharing information, for example, via shared electronic medical records was explicitly mentioned. It is of note that issues surrounding privacy were not mentioned by participants, although EMRs are in most cases not in place yet due to such issues at a wider system level.

Aspects especially to do with defining care seemed relatively unique and extra relevant for multimorbidity. However, there was a large degree of overlap in the concepts mentioned in defining health in the current focus groups and the existing definitions presented in the introduction by the WHO, Huber and Vankatapuram and similar studies conducted on defining health.[12–14] Namely, absence of shame, discrimination, fear and/or stigma; feeling safe; psychological well-being; and social relationships and participation especially overlap with the mental and social well-being aspects of the 1946 WHO definition of health.[12] The importance of mental health outcomes has also been recently found in a study among a UK general population on what aspects of the EuroQol 5-dimensions are currently missing or receive too little attention, that is, mental health.[20] This is also in line with a study by Ebrahimi and colleagues[21] among frail elderly where being able to 'master' daily life (eg, coping and acceptance) and being happy and satisfied with life (eg, enjoyment of life) were described as characteristics of health.[21]

Other concepts found in the current focus groups relate more to the definition by Huber and colleagues of health as the ability to adapt, that is, acceptance, coping and resilience.[13] Moreover, overlap was seen with our finding on the ability to conduct normal daily activities and the aforementioned study on frail elderly where the ability to live the routine life persons are accustomed to was identified as an aspect of health.[21] Lastly, Venkatapuram's definition of health as the capability to attain human value overlaps with the themes ability to participate in normal daily activities and social participation (a means to attain human value) and enjoyment of life that were mentioned in the current focus groups.[14]

The facets of good health found in the current study for a large part overlap with facets of functioning as set out by the International Classification of Functioning (ICF).[22] The goal of the ICF is to describe a person's functioning, irrespective of their specific health conditions, considering their context. The ICF can be used to classify aspects of functioning that can be measured when assessing the impact of, for example, integrated care programmes. However, in the current study, we aimed to identify aspects of (1) health and well-being and (2) experience with care that can be used when evaluating the impact of integrated care programmes on the Triple Aim. Our findings are reaffirmed by the fact that the different themes and concepts that persons in the focus groups used to define good health can be retraced to the ICF.

It was noticeable that, across all focus groups, aspects relating to care were already mentioned in defining good health. It appeared that persons could not always clearly distinguish the two. This might be due to the fact that persons more easily can identify these care-related factors as 'changeable'. When exploring which aspects of health/well-being and care were most important in part two of the focus groups, we saw across countries that these overlapped with those mentioned during the first part of the discussions but also that often concepts were included from the literature cards. As these were introduced and explained after the break, participants reacted positively to them. This indicates conceptual overlap in what is currently being assessed in evaluations, and thus found in the literature a priori, and what participants of our focus groups found important.[5]

## Strengths and limitations

Several strengths and potential limitations in the current study are described below. Strengths include the use of a broad set of inductively and deductively deduced outcomes, the variation between the countries in which the focus groups were held, the fact that persons with multimorbidity were so actively involved in this research and that the focus was on health and on the care process. Potential limitations may be the cross-country differences in discussing mental health problems, the time restriction that slightly changed the planned structure of the focus groups and the variation in specificity between concepts.

A difference in explicitly naming mental health problems between countries was observed. Culturally related stigma issues may play a role. In the UK, mental health problems were explicitly mentioned by participants when they listed their morbidities. This was not the case in the Netherlands, although throughout the focus group, it became apparent that depression issues were also present among participants. Also, in the German focus groups, mental health problems, such as depression, were mentioned as 'side effects' of other health problems. It is possible that our findings would have differed had more persons with (diagnosed) psychological multimorbidities been present at the focus groups. It could be that different aspects related to health and care would have been pointed out, for example, because in many healthcare and social care systems of the included eight countries there is fragmentation between psychological and somatic healthcare provisions. It is possible that certain findings would have been strengthened, such as the importance of being approached holistically and that somatic care professionals pay attention to psychological health problems and provide emotional support. Although having a more varied group of participants in this sense would be desirable, it is not a requirement nor a goal in qualitative studies to attain a representative sample.

A strength of the current study is that inductively and deductively derived outcomes were used in the second part of the focus groups. We chose to do so in order to provide participants with a broad set of possible outcomes to choose from. The added outcomes from the literature were, alongside novel concepts from part one of the focus groups, on participants' 'top 10' lists (see online supplementary file box S1). This indicates that simply because an outcome was not mentioned inductively in part one of the focus group, this did not mean that participants did not find it important.

Initially, the goal of the second part of the focus groups was to reach a group 'top 10', in which participants would discuss their individual lists and try to create a group list. However, due to time restrictions, this was not possible. Thus, the results of part two reflect each individual's opinion. We opted to present findings in the results section mentioned by 10 or more persons across the focus groups, but this is a relatively arbitrary cut-off point, and the full findings are shown in the supplementary file box S3. The total number of cards from which participants could select concepts for their top 10 list differed in size depending on from which country they are from. The level of specificity in concepts also differs greatly, for example, the overall satisfaction with care process is mentioned, as well as the specific facet that a provider has enough time to listen to a patient (see online supplementary file box S3). Quantifying such a qualitative process is thus to be done cautiously. In the online supplementary file box S3, concepts are clustered into broader categories. There are, however, different categorisations possible, since the concepts are all inter-related and different definitions can be applied.

We consider it a strength of the current research that the participating countries in SELFIE differ in geographical size, population (density), culture and welfare and healthcare systems, providing a wide scope. Furthermore, it is a main strength that persons with multimorbidity were so actively approached and included in the current study and in the wider SELFIE study, introduced in box 1. As will be described below, the results of these focus groups form the predominant input for the core set of outcomes to be included in the SELFIE evaluations of 17 integrated care programmes: participants were given a direct voice and greatly valued this. Another main strength of the current study and great addition to the literature is the fact that we asked participants in our focus groups to reflect on health and on experience with care. In situations whereby health improvements, especially in the traditional physical and clinical health indicators, are not feasible, focusing on satisfactory and even positive care process experiences becomes more important. Moreover, as was seen in the current study, experience with care is often inextricably linked to the definition of health. Although we did not explicitly incorporate costs in the first part of the focus groups when discussing what outcomes are important, participants did mention several cost concepts such as commuting costs and out-of-pocket fees. In the future, it would be interesting to also have an explicit discussion on what role costs play in, especially, the choice of and experiences with care.

### Next steps

When evaluating integrated care programmes for multimorbidity, a wide array of healthcare and care outcomes should be included. Few aspects related to physical functioning, a traditionally used outcome, were found on the top 10 lists of importance, that is, only 'energy and fatigue' was mentioned by more than 10 persons across all focus groups. Instead, more well-being, social and psychological facets of health are important and should thus be included in evaluations. This finding overlaps with current trends in conducting more comprehensive evaluations. To this end, in SELFIE, a core set of outcomes for the evaluation of 17 promising integrated care programmes for multimorbidity was developed that places a lot of focus on such outcomes. The core set consists of the following outcomes: physical functioning, psychological well-being, social relationships and participation, enjoyment of life, resilience, person-centredness, continuity of care and total healthcare and social care costs. This core set was developed on the basis of four sources, of which the focus groups described in the current article had the most prominent influence. The other three sources were: (1) findings from a literature review conducted to develop a framework on integrated care for multimorbidity and to identify existing programme evaluations,[5] (2) (inter) national SAB discussions in the eight SELFIE partner countries with patients, partners, professionals, payers and policy makers and (3) the aims of the aforementioned 17 programmes and current indicators included

in these programmes. A set of criteria was used to come to this list (see online supplementary file box S4), and some concepts mentioned in the focus group were not included in the core set, because these did not meet these criteria. This is the case for the more systemwide and cultural-wide concepts that cannot be used to assess the performance of one specific care programme, such as, for example, the absence of public stigma.

### CONCLUSION

In the current study, persons with multimorbidity in eight European countries described what good healthcare and care means to them. Beyond traditional outcomes, aspects such as acceptance, absence of shame and enjoyment of life came forth from the discussions. Especially important for persons with multimorbidity was to be approached holistically and to have continuity of care. Thus, a variety of health outcomes that entail well-being, social and psychological facets should be included when evaluating integrated care programmes for persons with multimorbidity. Even more imperative is that experience with care outcomes be included in such evaluations that tackle the complexity of multimorbidity care provision. In conclusion, important next steps include moving towards harmonising evaluation frameworks and the specific indicators used in such evaluations.

**Author affiliations**

[1]Erasmus School of Health Policy & Management, Erasmus University Rotterdam, Rotterdam, The Netherlands

[2]Department of Health Care Management, Berlin University of Technology, Berlin, Germany

[3]Institute for Advanced Studies, Vienna, Austria

[4]NIHR Greater Manchester Patient Safety Translational Research Centre, University of Manchester, Manchester, UK

[5]Syreon Research Institute, Budapest, Hungary

[6]Healthcare Financial Management Department, University of Pécs, Pécs, Hungary

[7]Uni Research Rokkan Centre, Bergen, Norway

[8]Consorci Institut D'Investigacions Biomediques August Pi i Sunyer (IDIBAPS), Barcelona, Spain

[9]Agency for Quality and Accreditation in Health Care and Social Welfare, Zagreb, Croatia

[10]Institute for Medical Technology Assessment, Erasmus University, Rotterdam, The Netherlands

**Acknowledgements** We would like to thank the focus group participants for making this research possible. We would also like to thank the entire SELFIE consortium for their support.

**Contributors** FL, MB, and MPMHR-vM drafted the focus group procedure document and analysis instruction document and organised and were present at the Dutch focus group. VS, MK, SC-S, AZ, RE, CV, MiH and TC were involved in the translation and organisation of the focus groups in their respective countries and in the analysis of their focus group and reporting thereof. FL, MB and MPMHR-vM analysed the Dutch group; FL, MaH, MB and MPMHR-vM analysed all focus group reports and conducted the overarching analyses. FL, MaH and MPMHR-vM drafted the manuscript and the revision; all coauthors critically reviewed and contributed to the manuscript and revision, for example, by appraising themes and providing fitting quotes.

**Funding** The SELFIE project has received funding from the European Union's Horizon 2020 research and innovation programme under grant agreement no. 634288.

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
