## [Reviewer comments · BMJ Open]

ARTICLE DETAILS

TITLE (PROVISIONAL)	Defining good health and care from the perspective of persons with multi-morbidity: Results from a qualitative study of focus groups in eight European countries
AUTHORS	Leijten, Fenna; Hoedemakers, Maaïke; Struckmann, Verena; Kraus, Markus; Cheraghi-Sohi, Sudeh; Zemplyni, Antal; Ervik, Rune; Vallve, Claudia; Huic, Mirjana; Czypionka, Thomas; Boland, Melinde; Rutten-van Mólken, Maureen

VERSION 1 – REVIEW

REVIEWER	Jan D. Reinhardt Sichuan University, China
REVIEW RETURNED	22-Jan-2018

GENERAL COMMENTS	The manuscript aims to explore the understanding of health and factors relevant to health care delivery from the perspective of people with multimorbidity. While these aims are relevant, the paper is well written and the methods are largely appropriate, I have one major and several minor concerns that should be addressed by the authors. Major issue: My major concern is that the authors completely ignore the literature on WHO's International Classification of Functioning, Disability and Health (ICF) and related research. It has been argued that with this classification the WHO provides an operationalization of their comprehensive understanding of health. Functioning and Disability in the ICF framework are understood as a comprehensive interplay of body functions/structures, activities and participation that in turn is influenced by health conditions, environmental factors and personal factors. Rijken et al. have for instance suggested to use an ICF-based approach in the management of patients with multimorbidity (https://www.ncbi.nlm.nih.gov/pubmed/22712877). I believe that considering this model would strengthen the theoretical approach of the authors and ease the interpretation of findings, particularly in relation to the patients' understanding of health. Moreover, based on ICF various so called ICF Core Sets as selections of ICF categories (i.e. health and health-related aspects) of major relevance to people with particular health conditions and in particular settings have been developed, among others based on focus groups with persons having particular health problems (see Selb et al. 2015 for an overview: https://www.ncbi.nlm.nih.gov/pubmed/24686893). In addition, ICF-linking has been used in various qualitative studies aiming at understanding people's perspective on participation, health-related
--

	quality of life and the like. This provides a lot of literature the authors could compare their findings with (Comparison with other studies is not well developed in the Discussion). ICF linking (see Cieza et al. 2016: https://www.ncbi.nlm.nih.gov/pubmed/26984720) is furthermore a method the authors may consider in analysing their findings. Minor issues:  1. Line 89: “reducing cost [increase]”; I don’t understand what is meant by “increase”, certainly not increase the costs. 2. Lines 100-102: I wish the authors could reformulate their objective and specific aims. To “have persons with multi-morbidity define good health or well-being and a good care process” is not an aim but actually what the authors do in the study. An aim would be, for instance, “to better understand the meaning of good health from the perspective of people with multimorbidity”. 3. Please clarify the recruitment procedure. Were people with multimorbidity from the stakeholder boards contacted and then the snowball procedure used and the people recruited this way were from different organizations such as advocacy groups or were organizations such as advocacy groups contacted independently or both? 4. Ethics approval: My understanding is that this study involved people with multimorbidity living in the community and not inpatients and thus was exempt from Institutional Review Board ethical approval. Am I correct? If so, please mention. Please also mention that the study was conducted according to the principles of the Helsinki Declaration. 5. Given above, you may consider not to speak of “patients” but persons with multimorbidity or multiple health conditions.
--	--

REVIEWER	Kerry Kuluski Lunenfeld--Tanenbaum Research Institute, Sinai Health System and University of Toronto, Canada.
REVIEW RETURNED	03-Feb-2018

GENERAL COMMENTS	This is a really interesting and important paper about what is most important to people in relation to their health and care across multiple European countries. I have a number of suggestions that I think will strengthen the manuscript. I feel that the background lacks an overview of the current empirical literature in relation to the paper topic. Clearly the authors understand the breadth of literature on their topic (given that existing outcomes of importance are outlined in Box S.2- Pages 32 and 33 and were used in the focus groups). Can these be woven into the background to situate the work? Also, not sure why there is so much emphasis on QALYs when so many other outcomes have been measured in previous studies. Can the authors describe why they brought in concepts from the literature instead of just using the concepts shared in the focus groups? What strengths and limitations does this approach have? Once everything was written on note cards (from part 1 and the literature) did participants have an opportunity to introduce new ideas at that time? Were these incorporated? More information is needed in the analysis section. It is unclear to me how exactly the 'inductive' and 'deductive' analysis was
--

	conducted. Given the multiple languages across focus groups, how were data translated to English and how did authors reach consensus on core concepts? The data as presented seem to be categories with a range of examples, and not themes. Why did the authors not attempt to conceptually group the content? I find that some grouping of content would make it easier to digest the information. The authors talk about generalizability but this is not appropriate language for a qualitative study. The term transferability should be used instead and the extent to which their data are transferable to other populations with multi morbidity. The authors justify use of focus groups as appropriate to allow "people with multimorbidity to have a clear voice" -- this is oddly worded and seems like a bit of an assumption. Might be more appropriate to state something like: the focus group method allowed the researcher to gather perspectives from multiple people on a given topic. A skilled facilitator made efforts to ensure that everyone's perspective of shared. Here are some more specific comments: Page 7, Line 134- " a gratification was made available" this sounds odd to me-- are you referring to a gift card, compensation? Page 8, Line 171- why did the facilitators ask "how can you expand this so it counts for everyone in the room?" I don't really understand this question. Do you mean so it's understood by everyone in the room? Why did the authors choose to use a priori outcomes? Page 11- lines 227 and 228- wording issue- "only in some cases specific persons not participate" - add "did" before specific Page 14- Line 348- "professionals does do not focus" - remove word "does" page 16- Line 402- what do the authors mean by "time" Line 415- "several themes were only brought up in Germany and Netherlands"-- as I read this I wonder if these are actually themes? They read like categories or examples. Line 453- Change to- "The following health and well-being concepts were most frequently noted by participants in the top 10 lists": Page 18, line 462- do you mean "within and between" countries? Page 19, lines 486 and 487- make the examples more specific in brackets. Page 20- at the end of discussion can you comment on what stood out in your data? What did the study add, over and above what is known in the literature? Did anything surprise you? Page 21- line 543- remove the term representativeness and comment on transferability (see my earlier comment).
--	--

	Page 22- line 600- remove the word *is* Was institutional ethics required? I only see a comment on consent forms. The appendices look great- thank you for outlining your focus group approach/guides so clearly. In the Boxes that outline all the concepts -- is there a way to group them so they are easier to digest? The top 10 lists (and the frequencies) makes me think that a summative content analysis should be outlined as part of your method. I also noticed that some of the bolded terms (which appear to be headings have frequencies assigned while other do not...typo?) Overall, I enjoyed reading this paper and found it very interesting and relevant. The above comments, in my view, will strengthen the paper (particularly the methods and presentation of data).
--	--

REVIEWER	Kate O'Loughlin The University of Sydney, Australia
REVIEW RETURNED	21-Feb-2018

GENERAL COMMENTS	Overall this paper makes a contribution to the research looking at integrated care systems, and through the use of one focus group per country provides a glimpse of what is happening in the eight countries included. I support the publication of the paper but believe it needs a little more work. I have made specific comments and suggestions in the attached file - these are mainly around providing more context for aspects of the study and including additional or clarifying details. As it seems there are multiple authors/contributors, careful proofreading is required to ensure consistency and clarity of expression through the paper. - The reviewer also provided a marked copy with additional comments. Please contact the publisher for full details.
---

VERSION 1 – AUTHOR RESPONSE

Reviewer 1 comments:

Major points:

1. My major concern is that the authors completely ignore the literature on WHO's International Classification of Functioning, Disability and Health (ICF) and related research. It has been argued that with this classification the WHO provides an operationalization of their comprehensive understanding of health. Functioning and Disability in the ICF framework are understood as a comprehensive interplay of body functions/structures, activities and participation that in turn is influenced by health conditions, environmental factors and personal factors. Rijken et al. have for instance suggested to use an ICF-based approach in the management of patients with multimorbidity (<https://www.ncbi.nlm.nih.gov/pubmed/22712877>).

I believe that considering this model would strengthen the theoretical approach of the authors and ease the interpretation of findings, particularly in relation to the patients' understanding of health.

Moreover, based on ICF various so called ICF Core Sets as selections of ICF categories (i.e. health and health-related aspects) of major relevance to people with particular health conditions and in particular settings have been developed, among others based on focus groups with persons having particular health problems (see Selb et al. 2015 for an overview: <https://www.ncbi.nlm.nih.gov/pubmed/24686893>). In addition, ICF-linking has been used in various qualitative studies aiming at understanding people's perspective on participation, health-related quality of life and the like. This provides a lot of literature the authors could compare their findings with (Comparison with other studies is not well developed in the Discussion).

ICF linking (see Cieza et al. 2016: <https://www.ncbi.nlm.nih.gov/pubmed/26984720>) is furthermore a method the authors may consider in analysing their findings.

Reply: We agree that the ICF is an important classification. There is certainly overlap in the facets of functioning that are defined in the ICF and facets of defining good health found in our study. However, our study is predominantly framed in terms of outcomes that should be measured when assessing the impact of integrated care programmes on the Triple Aim. Improving population health is just one of the Triple Aims, whereas the other two are improving patient's experience with care and reducing costs (not included in the ICF). Functioning and health are closely linked, and it is thus reassuring that there is overlap between our health-findings and facets of functioning in the ICF. When looking at the general ICF Core Set, there is some overlap with our main findings. However, themes and concepts that arose from our study such as acceptance of a disease, feeling safe, stigma and shame, coping, and being able to enjoy life are not included in the ICF general core set. We have now described the overlap and differences in our findings with the ICF.

New text (Discussion, p 20-21, lines 568-576):

The facets of good health found in the current study for a large part overlap with facets of functioning as set out by the International Classification of Functioning (ICF) (WHO, 2001). The goal of the ICF is to describe a person's functioning, irrespective of their specific health conditions, considering their context. The ICF can be used to classify aspects of functioning that can be measured when assessing the impact of, for example, integrated care programmes. However, in the current study we aimed to identify aspects of 1) health and wellbeing, and 2) experience with care, that can be used when evaluating the impact of integrated care programmes on the Triple Aim. Our findings are reaffirmed by the fact that the different themes and concepts that persons in the focus groups used to define good health can be retraced to the ICF.

Minor points:

2. Line 89: "reducing cost [increase]"; I don't understand what is meant by "increase", certainly not increase the costs.

Reply: What is meant here is to not only reduce the costs in absolute terms but also the (annual) growth in costs over time (the upwards trend). We've replaced the word 'increase' by the word '-growth'.

3. Lines 100-102: I wish the authors could reformulate their objective and specific aims. To "have persons with multi-morbidity define good health or well-being and a good care process" is not

an aim but actually what the authors do in the study. An aim would be, for instance, “to better understand the meaning of good health from the perspective of people with multimorbidity”.

Reply: We have changed this.

Changed text (Introduction, p 5, lines 112-115):

In the current study we aim to better understand what good health or well-being and a good care process mean from the perspective of persons with multi-morbidity and to identify what they find most important in each. We hereby thus focus on two of the three ‘Triple Aims’.

4. Please clarify the recruitment procedure. Were people with multimorbidity from the stakeholder boards contacted and then the snowball procedure used and the people recruited this way were from different organizations such as advocacy groups or were organizations such as advocacy groups contacted independently or both?

Reply: Persons were recruited in different ways. The response to the above question is thus ‘both’. See changed text below.

Changed text (Methods, p 6, lines 142-143 & lines 151-153):

Different recruitment strategies were applied: participants were recruited via patient organisations (AT, DE, HR, NL, NO), medical professional organisations (HR), ... These persons [SAB members] were reached out to for the focus groups and we tried to snowball via their networks, e.g., also via patient organisations.

5. Ethics approval: My understanding is that this study involved people with multimorbidity living in the community and not inpatients and thus was exempt from Institutional Review Board ethical approval. Am I correct? If so, please mention. Please also mention that the study was conducted according to the principles of the Helsinki Declaration.

Reply: See response to Editor’s comment 4.

6. Given above, you may consider not to speak of “patients” but persons with multimorbidity or multiple health conditions.

Reply: We have now do not use the word “patients” when referring to the participants, and only use the word when describing the theme of provider-patient relations, because this pertains to a care setting in which these are the roles between the two involved persons.

Reviewer 2

Major points:

1. I feel that the background lacks an overview of the current empirical literature in relation to the paper topic. Clearly the authors understand the breadth of literature on their topic (given that existing outcomes of importance are outlined in Box S.2- Pages 32 and 33 and were used in the focus groups). Can these be woven into the background to situate the work?

Reply: Important elements of integrated care for multi-morbidity were identified and summarised in the publication by Struckmann et al., 2017. The publications included for this review were also used to inform the background section in the current manuscript. When these publications were integrated care programme evaluations, they were used to make the a priori cards for part two of the focus groups. We have added information on this.

New text (Introduction, p 5, lines 105-110):

Scientific literature found for a recent review also conducted in the context of the SELFIE research project, as the current study is (see Box 1), provided insight into which outcomes have been used in recent integrated care programme evaluations (Struckmann et al., 2017). We saw that a wide array of indicators corresponding to the Triple Aim are being used. This is also reiterated by the findings in the review by Linton et al. (2016), whereby a total of 99 instruments to assess well-being were found, with a great variety between these (Linton et al., 2016).

Changed text (Methods, p 8, lines 197-201):

The outcomes on the a priori made cards stemmed from publications on integrated care programmes for multi-morbidity that were included in a large scientific literature review that was also conducted in the SELFIE project (Struckmann et al., 2017).

2. Also, not sure why there is so much emphasis on QALYs when so many other outcomes have been measured in previous studies.

Reply: The emphasis in the Introduction on QALYs stems from the call to use of cost-effectiveness studies to inform the decision-making on the continuation, wider implementation, or reimbursement of care programmes. We have added a sentence in the Introduction to make the link between the decision-making context and the use and limitation of QALYs clearer.

Changed text (Introduction, p 5, lines 78-83):

Increasing this evidence-base is important for the durability, wider implementation, and more sustainable reimbursement/financing of such programmes. Often, decisions on these matters are informed by economic evaluations in which costs per Quality Adjusted Life Years (QALYs) are

calculated. However, it can be questioned whether the current (economic) evaluation framework provides sufficient insight into the broad range of outcomes that such integrated care programmes aim to improve.

3. Can the authors describe why they brought in concepts from the literature instead of just using the concepts shared in the focus groups? What strengths and limitations does this approach have? Once everything was written on note cards (from part 1 and the literature) did participants have an opportunity to introduce new ideas at that time? Were these incorporated?

Reply: We chose to incorporate both concepts from the first part of the focus group and from the literature in the second part of the focus group in order to provide participants with a large overview of possible outcomes. In order to make the advantages of doing so clearer in the manuscript, we have added explanation on this in the Methods and Discussion sections.

New text (Methods, p 8, lines 204-206):

The purpose of including also these outcomes in the second part of the focus groups was to provide participants with a large and 'complete' (on the basis of the available literature) overview of possible outcomes.

New text (Discussion, p 21-22, lines 604-610):

A strength of the current study is that inductively and deductively derived outcomes were used in the second part of the focus groups. We chose to do so in order to provide participants with a broad set of possible outcomes to choose from. The added outcomes from the literature were, alongside novel concepts from part one of the focus groups, on participants' 'top-10' lists [...]. This indicates that simply because an outcome was not mentioned inductively in part one of the focus group, this did not mean that participants did not find it important.

4. More information is needed in the analysis section. It is unclear to me how exactly the 'inductive' and 'deductive' analysis was conducted.

Reply: We have expanded upon the description of our analyses in the Methods.

Changed text (Methods, p 9, lines 229-235):

The analysis of part one was done applying an inductive (bottom-up) approach, whereby different specific concepts mentioned by participants were analysed by describing these, looking at their overlap and differences, and when possible clustering these into more general concepts. By deciding upfront that the procedure and our findings would be structured according to the Triple Aim, we also applied a deductive (top-down) approach. For part two, further clustering was done to present the findings in a clearer way, and a summation was used to identify concepts that were found most important.

5. Given the multiple languages across focus groups, how were data translated to English and how did authors reach consensus on core concepts? The data as presented seem to be categories with a range of examples, and not themes. Why did the authors not attempt to conceptually group the content? I find that some grouping of content would make it easier to digest the information.

Reply: Researchers, present during their respective focus groups, analysed their notes and when necessary recordings (in their respective language) and provided an English report to the Dutch team. The quotes presented throughout the manuscript were translated to English by these same researchers. We have clarified this in the Methods section. Throughout the analyses process (see also response 4 above) outcomes were conceptually clustered, with the overarching categorisation being the Triple Aim. By using bold lettering in the Results section, we draw attention to larger concept-clusters of outcomes. This clustering can also be seen in Supplementary File Box S.3.

Changed text (Methods, p 8, lines 217-221):

Reports were made in English by each SELFIE partner on their respective focus group [...].

6. The authors talk about generalizability but this is not appropriate language for a qualitative study. The term transferability should be used instead and the extent to which their data are transferable to other populations with multi morbidity.

Reply: See response to Editor's comment 2.

7. The authors justify use of focus groups as appropriate to allow "people with multimorbidity to have a clear voice" -- this is oddly worded and seems like a bit of an assumption. Might be more appropriate to state something like: the focus group method allowed the researcher to gather perspectives from multiple people on a given topic. A skilled facilitator made efforts to ensure that everyone's perspective of shared.

Reply: See response to Editor's comment 2.

Minor points

8. All minor grammatical points have been addressed.

9. Page 20- at the end of discussion can you comment on what stood out in your data? What did the study add, over and above what is known in the literature? Did anything surprise you?

Reply: We have added two sentences in the Conclusion to highlight several of the most interesting points.

New text (Discussion, p 23, lines 665-668):

Beyond traditional outcomes, aspects such as acceptance, absence of shame, and enjoyment of life came forth from the discussions. Especially important for persons with multi-morbidity was to be approached holistically and to have continuity of care.

10. Page 21- line 543- remove the term representativeness and comment on transferability (see my earlier comment).

Reply: The paragraph on generalizability and representativeness in the Discussion has been removed. We have also added a sentence on this.

New text (Discussion, Strengths & limitations, p 21, lines 602-603):

Although having a more varied group of participants in this sense would be desirable, it is not a requirement nor a goal in qualitative studies to attain a representative sample.

11. Was institutional ethics required? I only see a comment on consent forms.

Reply: See response to Editor's comment 4.

12. In the Boxes that outline all the concepts -- is there a way to group them so they are easier to digest?

Reply: Please see the response to comment 5. In Supplementary File Box S.3 the concepts were clustered along the lines used in the Results section of the main text.

13. The top 10 lists (and the frequencies) makes me think that a summative content analysis should be outlined as part of your method. I also noticed that some of the bolded terms (which appear to be headings have frequencies assigned while other do not...typo?)

Reply: See responses to comments 4 & 5 above.

Reviewer 3

Comments:

1. Box 1 SELFIE: While the explanation of the SELFIE project is helpful, as a reader and not from one of the eight countries or from anywhere else in Europe I'm at a bit of a loss to understand the context of the research. Do these countries have similar health and social care structures/provisions? Do they have formalised and comparable integrated care provisions for those with multi-morbidity based on EU guidelines? There is a general statement about different programmes being implemented across Europe but nothing specific about any of the eight countries -

even a brief statement about this group of eight would provide a context for the FGs that were carried out.

Reply: All of these countries have a long history of universal health care systems, either tax-funded or funded by insurance premiums. However, these countries are not similar when it comes to their health- and social care provider systems. We have added information on this in the Methods section of the manuscript and in the Box on SELFIE.

New text (Methods, p 6, lines 130-135):

These countries have a long history of universal health care systems that are either tax-funded or funded through insurance premiums. However, they each have a unique health- and social care provider system in which persons with multi-morbidity might face different challenges. For more information on the organisation of care in each context, see the macro level descriptions of each country in the 'thick description' reports available on the SELFIE website [...].

Changed text (Box 1, p 27):

The SELFIE consortium includes eight organisations in the following countries [...]. Each country has a unique health- and social care system, that varies in the extent to which health- and social care are integrated and that vary in how the financing is arranged. On the SELFIE website [...] a macro level description of the systems can be found that provides background context.

2. Methods, Procedure: This is quite an elaborate recruitment process for procuring one FG in each country! Could be written up in a more concise manner?

Reply: We applied the COREQ guidelines for reporting on qualitative research. While keeping to these, we have shortened the Procedure section where possible.

3. Methods, Procedure: The presence of four researchers in one group of 6-8 participants is potentially intimidating and a generous use of resources.

Reply: The upside of having more researchers present was that multiple persons could take extensive notes, thus providing for more materials and discussants for the analyses. The potential downside of intimidating participants, was something we tried to deal with by encouraging an informal introduction by the researchers. Please also note that in the focus group where four researchers were present, two solely took notes.

4. Methods, Procedure: Purpose of the 'extensive notes' if the FGs recorded? Are the notes included in the analysis/interpretation of the findings?

Reply: The extensive notes were used in analysing and interpreting findings. The recordings were not transcribed verbatim (in most cases) and not the basis of each country-report. The recordings were used to check findings and for quotes. Please see the section Analyses in the Methods for more information.

5. Methods, Procedure: How was consensus reached on 'top 10'?

Reply: Consensus was not reached on the 'top 10' lists in part two of the focus groups. The concepts that we present in Table 3 are the ones that were included the top-10 lists of at least 10 persons.

6. Methods, Procedure: This procedure sounds more like a work-shopping exercise rather than a focus group! What was the expected duration of the FG? Were participants given a time frame? FGs conducted in language of each country? Materials/concepts translated into each language?

Reply: We understand that using flip-overs and notecards is something that is often done in workshops. However, the procedure followed throughout the entire session was that of a focus group, whereby participants interacted primarily with one-another and not with the researchers/organisers. The expected duration of the focus groups was two hours. Each focus group was conducted in the respective language of the country it was being held in. The materials, i.e., the a priori determined outcomes on notecards, were translated by each partner into their respective language.

New text (Methods, p 7, line 169):

The planned duration of the focus groups was two hours.

New text (Methods, p 7, lines 159-160):

Each country made a protocol in their own language to use during the actual focus group, which was held in their respective language.

7. Results, Defining health and care (part 1), Defining care: Regarding self-help groups: For individual disease(s) or multi-morbidity? Were participants asked if they belonged to self-help groups?

Reply: In Austria the role of self-help groups was mentioned by participants several times as a means of getting support and learning how to deal with problems. We did not ask participants whether they were members of a self-help group, however, some mentioned this themselves in the introduction rounds at the start of the focus group.

8. Results, Defining health and care (part 1), Defining care: Who is the 'provider' in this case? Health professional? Hospital? Social care provider? Case manager? Not clear whether there is one provider addressing all health and social care needs based on multi-morbidity.

Reply: The word 'provider' can pertain to both a care professional and a care organization; we have thus changed this into the word professional. Between and within countries there is variation in who the professional is, depending on the individual's specific multi-morbidities and the health- and social care system in their respective country. Regardless of which specific professional is the main contact point / care provider, the finding pertains to the importance that there is a relationship of trust between the two. Also see the reply to comment 9 below.

9. Results (Table 3): I'm unclear on who is seen as providing 'care' in the findings reported here. From the Introduction to this paper, I understood the study was intended to elicit views and experiences related to health and social care for participants with multi-morbidities, however what is reported here seems to be referring only to medical care provided by doctors/physicians and a more generic 'provider'. For me, social care relates to community and home care needed to support people, but this does not seem to figure in the findings. Who is expected to provide psychological and emotional support? The GP? Who should be responsible for continuity of care? I refer back to my earlier point: do these countries have similar health and social care structures/provisions?

Reply: The care professionals referred to by participants are both health- and social care professionals. The emotional and psychological support that these persons with multi-morbidity find important needs to come from their care professional, regardless of their specific expertise. It's important that all types of care professionals pay attention to the emotional and psychological factors. Please see the new text added on this in the Discussion.

New text (Discussion, p 19, lines 534-541):

Considering the different combinations of morbidities represented in the focus groups and the fact that participants are living in different health- and social care contexts, it is interesting to note that regardless of whether the main care professional came from primary, secondary, or social care, these aspects relating to respect, providing emotional support, and expertise were important for many participants. This might be central to the fact that persons with multi-morbidity deal with multiple care professionals who need to have attention for the person as a whole, considering all their morbidities and the emotional burden that comes alongside these.

10. Discussion, Main findings: Were participants in each country's FG drawn from a major metropolitan centre and/or regional/rural settings? If access to services is dependent on geography, how/why did this come up in the FG discussion?

Reply: Most of the focus groups were held in metropolitan areas (see Editorial Requests comment 3 above), however, participants were not always living in these cities and commuted for the purpose of the focus groups. Participants themselves noted the issue of geographical access in the focus groups in Norway, Spain and UK. This could be related to the fact that services are more spread out in these,

relatively larger, countries. However, since this is a qualitative study participants are not representative for their country.

11. Discussion, Strengths & limitations: I do not see identifying having a mental health issue as one of a participant's multi-morbidities is the same as identifying the need for psychological and emotional support for people living with multiple morbidities - please clarify.

Reply: We agree that these two things, explicitly naming mental health problems and identifying the need for emotional support, are not necessarily linked. We have re-written this paragraph.

Changed text (Discussion, Strengths & limitations, p 21, lines 589-603):

Only in the UK mental health problems were explicitly mentioned by participants when they listed their morbidities at the start of the focus groups. The difference in explicitly naming mental health problems between countries may relate to culturally-related stigma issues. For example, throughout the Dutch focus group it became apparent that depression issues were also present amongst participants. Also in the German focus groups mental health problems, such as depression, were mentioned as 'side effects' of other health problems. It is possible that our findings would have differed had more persons with (diagnosed) psychological multi-morbidities been present at the focus groups. It could be that different aspects related to health and care would have been pointed out, for example because in many health- and social care systems of the included eight countries there is fragmentation between psychological- and somatic health care provisions. It is possible that certain findings would have been strengthened, such as the importance of being approached holistically and that somatic care professionals pay attention to psychological health problems and provide emotional support. Although having a more varied group of participants in this sense would be desirable, it is not a requirement nor a goal in qualitative studies to attain a representative sample.

12. Discussion, Strengths & limitations: Regarding costs: in prioritising what?? Costs associated with health and care? Personal costs? System costs? As we don't know anything about the health and care systems in these countries, bringing up costs here does seem very relevant

Reply: We refer to costs in evaluating integrated care programmes and in including them as outcomes in such evaluations. Costs can include the different categories mentioned by the reviewer. We mention this in the limitations because we focused on two of the three Triple Aims (health and experience) and did not ask participants what types of cost outcomes they find important. We've rewritten the sentences on this to clarify and provide examples.

Changed text (Discussion, p 22, lines 636-639):

Although we did not explicitly incorporate costs in the first part of the focus groups when discussing what outcomes are important, participants did mention several cost concepts such as commuting costs and out-of-pocket fees. In the future, it would be interesting to also have an explicit discussion on what role costs play in, especially, the choice of and experiences with care.

13. All minor

VERSION 2 – REVIEW

REVIEWER	Jan D. Reinhardt Institute for Disaster Management and Reconstruction, Sichuan University and Hong Kong Polytechnic University, Chengdu, China.
REVIEW RETURNED	24-Apr-2018

GENERAL COMMENTS	The absence of a rebuttal letter made this re-review a little difficult. I also do not know what the comments of the other reviewers were. My own comments have however been sufficiently addressed by the authors. There are only minor issues remaining: 1. In the abstract, I would suggest to also reformulate the objective such as at the end of the introduction.2. I would also like to suggest to better balance the strength and limitations part. Currently, there is one strength and three limitations.3. The first limitation mentioned, i.e. that only in the UK mental health problems were mentioned and that the population with mental health problems may have been under-represented, does not appropriately reflect the more complex picture on this issue provided in the discussion.
---

VERSION 2 – AUTHOR RESPONSE

Comments and replies:

1. In the abstract, I would suggest to also reformulate the objective such as at the end of the introduction.

The objective in the abstract has been changed.

New text: Focus groups were held with persons with multi-morbidity in eight European countries to better understand what good health and a good care process means to them and to identify what they find most important in each.

Old text: To gain insight into meaningful outcomes, focus groups were held with persons with multi-morbidity in eight European countries, in which participants defined 'good health and well-being' and a 'good care process'.

2. I would also like to suggest to better balance the strength and limitations part. Currently, there is one strength and three limitations.

We have added several sentences at the start of the "Strengths & Limitations" section of the Discussion to make explicit what the strengths and potential limitations are before going into more detail into each. We feel that this makes the balance clearer.

New text: Several strengths and potential limitations in the current study are described below. Strengths include the use of a broad set of inductively- and deductively deduced outcomes, the variation between the countries in which the focus groups were held, the fact that persons with multi-morbidity were so actively involved in this research, and that the focus not only on health but also on

the care process. Potential limitations may be the cross-country differences in discussing mental health problems, the time restriction that slightly changed the planned structure of the focus groups, and the variation in specificity between concepts.

3. The first limitation mentioned, i.e. that only in the UK mental health problems were mentioned and that the population with mental health problems may have been under-represented, does not appropriately reflect the more complex picture on this issue provided in the discussion.

We have nuanced this description to focus less on the explicitly naming in the UK and more on general cultural differences. See also the reply above.

New text: A difference in explicitly naming mental health problems between countries was observed. Culturally-related stigma issues may play a role. In the UK mental health problems were explicitly mentioned by participants when they listed their morbidities. This was not the case in the Netherlands, although throughout the focus group it became apparent

Old text: Only in the UK mental health problems were explicitly mentioned by participants when they listed their morbidities at the start of the focus groups. The difference in explicitly naming mental health problems between countries may relate to culturally-related stigma issues. For example, throughout the Dutch focus group it became apparent